# Structure of the transporter associated with antigen processing trapped by herpes simplex virus

**Michael L Oldham[1], Nikolaus Grigorieff[2], Jue Chen[1]***

[1]Howard Hughes Medical Institute, The Rockefeller University, New York, United States; [2]Janelia Research Campus, Howard Hughes Medical Institute, Ashburn, United States

**Abstract** The transporter associated with antigen processing (TAP) is an ATP-binding cassette (ABC) transporter essential to cellular immunity against viral infection. Some persistent viruses have evolved strategies to inhibit TAP so that they may go undetected by the immune system. The herpes simplex virus for example evades immune surveillance by blocking peptide transport with a small viral protein ICP47. In this study, we determined the structure of human TAP bound to ICP47 by electron cryo-microscopy (cryo-EM) to 4.0 Å. The structure shows that ICP47 traps TAP in an inactive conformation distinct from the normal transport cycle. The specificity and potency of ICP47 inhibition result from contacts between the tip of the helical hairpin and the apex of the transmembrane cavity. This work provides a clear molecular description of immune evasion by a persistent virus. It also establishes the molecular structure of TAP to facilitate mechanistic studies of the antigen presentation process.

## Introduction

*For correspondence: juechen@rockefeller.edu

Cytotoxic T cells detect and eliminate infected cells by recognizing viral peptides displayed on the cell surface by major histocompatibility complex (MHC-I) molecules (*Blum et al., 2013*). The viral peptides are generated in the cytosol and loaded onto MHC-I molecules in the endoplasmic reticulum (ER). The transporter associated with antigen processing (TAP) transports these cytosolic peptides into the ER lumen, where a multi-component peptide-loading complex facilitates peptide-binding to nascent MHC-I molecules (*Neefjes et al., 1993*; *Shepherd et al., 1993*). Upon formation of a stable complex with peptides, MHC-I molecules are released from the ER and exported to the cell surface. Receptors on circulating T cells react to pathogen-derived and malignant peptides, leading to a cytotoxic event that kills the diseased cells. T cells that recognize peptides derived from normal cellular proteins are eliminated or inactivated during development to prevent an autoimmune response, a process called immune tolerance. TAP-deficient cells have a reduced surface expression of MHC-I molecules and are less sensitive to cytotoxic T cells (*Deverson et al., 1990*; *Monaco et al., 1990*; *Spies et al., 1990*; *Trowsdale et al., 1990*)

TAP is an ER-resident transporter formed by two homologous subunits, TAP1 and TAP2. Similar to other ABC transporters, it contains two nucleotide-binding domains (NBDs) that hydrolyze ATP and two transmembrane domains (TMDs) that bind the substrate. In addition, both TAP1 and TAP2 contain an N-terminal transmembrane region (TMD0) that interacts with tapasin to form the larger peptide-loading complex (*Hulpke et al., 2012*; *Procko et al., 2005*). TAP's broad substrate specificity is one of the most important properties for its function (*Androlewicz and Cresswell, 1994*; *Momburg et al., 1994a*; *Neefjes et al., 1995*; *van Endert et al., 1995*). Humans typically express six different MHC-I molecules, each binding to a large variety of peptides and conferring a different

specificity (*Falk et al., 1991*; *Hunt et al., 1992*; *Jardetzky et al., 1991*). The same TAP transporter provides peptides for all six MHC-I molecules; therefore, it must be more promiscuous than any single MHC-I molecule. The substrate specificity of TAP has been studied extensively. The only sequence constraint found for human TAP is a preference for a hydrophobic or basic residue at the C-terminus (*Momburg et al., 1994b*). Interestingly, this peptide preference complements MHC-I specificity in that an acidic C-terminus is rarely seen in MHC-I presented peptides (*Rammensee et al., 1999*).

Despite its ability to transport a diverse range of peptides up to 40 residues long (*Momburg et al., 1994a*), TAP is still subject to inhibition by some viral peptides. For example, human herpes simplex virus (HSV) encodes the potent TAP inhibitor ICP47 (*Früh et al., 1995*; *Hill et al., 1995*). Peptides containing the N-terminal 34 residues of ICP47 are sufficient to bind TAP and prevent peptide translocation (*Galocha et al., 1997*; *Neumann et al., 1997*). Suppressing the presentation of viral peptides renders HSV-infected cells undetectable to cytotoxic T cells (*York et al., 1994*). This mechanism contributes to the lifelong infection of HSV. It also raises an intriguing question: How does a viral peptide inhibit a promiscuous peptide transporter?

Previously, we determined the structure of a TAP/ICP47 complex using cryo-electron microscopy (cryo-EM) (*Oldham et al., 2016*). This structure, at 6.5 Å resolution, showed that the N-terminal region of ICP47 forms a helical hairpin, inserting itself into TAP's substrate translocation pathway. We have continued to study this system to establish the molecular structure of TAP and to understand the chemical nature of the inhibition. Here we present a 4.0 Å cryo-EM reconstruction of the human TAP/ICP47 complex and describe the specific atomic interactions that allow the viral peptide to bind tightly to TAP and lock it in an inactive state.

## Results and discussion

### Cryo-EM attempts to improve the resolution

One major modification we incorporated for our new data collection was using a higher magnification. As the ordered region of the TAP/ICP47 complex has a molecular mass of only 130 kDa and exhibits a pseudo-twofold symmetry, two factors likely limiting resolution are the accuracy with which the noisy cryo-EM images of single particles can be aligned and the ability to distinguish pseudo-symmetrically related views. Since the predominant secondary structure observed in TAP/ICP47 complex is helical, we expect that the signal in the 7–8 Å resolution range is important for alignment. The higher magnification reduced the effective pixel size of the images from 1.35 Å used in our previous study to 1.04 Å (*Table 1*), boosting the detective quantum efficiency (DQE) of the K2 Summit detector (Gatan, Inc.) at 7 Å resolution by about 6% (*Li et al., 2013*; *McMullan et al., 2009*; *Ruskin et al., 2013*). We also lowered the dose rate from 10 electrons per pixel to eight in order to minimize coincidence loss (*Li et al., 2013*) and limited data collection to areas of the grids that showed good contrast of the particles, presumably correlating with regions of the grids that had the thinnest ice.

To optimize data processing, we compared three methods to correct electron beam-induced specimen movement: whole frame alignment using Unblur (*Grant and Grigorieff, 2015*), individual particle alignment using alignparts_lmbfgs (*Rubinstein and Brubaker, 2015*), and a combination of both. The best result was obtained by whole frame alignment followed by individual particle tracking. This combination of methods also works best for other small particles such as the γ-secretase (*Bai et al., 2015*).

Reconstruction and refinement were performed in Frealign (*Grigorieff, 2016*) using the previous 6.5 Å reconstruction as an initial reference. As Frealign automatically weights each particle according to its correlation to the model, we obtained the best reconstruction without prior 2D or 3D classification using only Frealign's score-based particle weighting (*Grigorieff, 1998*). This resulted in weighting coefficients at 4 Å that varied by about ±10% (about 5% of the particles received such low scores that they were effectively excluded at 4 Å resolution). Hence, we used all 502,000 particles that were automatically extracted from the micrographs for the 3D reconstruction. To exclude variable density corresponding to the detergent micelles and the flexible TMD0 domains, alignment was performed with a mask surrounding only the structured region of the protein. This mask was generated from the 6.5 Å reconstruction and low-pass filtered to 8 Å to remove high-resolution information. The

**Table 1.** Summary of Cryo-EM data.

| Imaging | |
|---|---|
| Microscope | Titan Krios I, 300keV (FEI) |
| Detector | K2 Summit direct electron detector (Gatan) |
| Energy filter | 10 eV (Gatan) |
| **Data collection** | |
| Pixel size | 1.04 Å |
| Movies | 3875 |
| Frames | 50 |
| Total exposure time | 10 s |
| Exposure time per frame | 0.2 s |
| Total exposure | 74 electrons/Å$^2$ |
| Exposure per frame | 1.48 electrons/Å$^2$/frame |
| Defocus range | −1.5 to −3.5 μm |
| **Final reconstruction** | |
| Number of particles | 501,973 |
| B-factor correction | −150 Å$^2$ |

mask was then applied using Frealign's 3D masking option (*Grigorieff, 2016*), specifying a smooth edge of about 5 Å and leaving the density outside the mask in place and low-pass-filtered to 30 Å resolution. This enabled parts of the disordered regions of the complex to contribute to the alignment at low resolution while only the signal from the well-ordered parts of the complex contribute at high resolution.

The final reconstruction has an average resolution of 4.0 Å (*Figure 1A and B*). Analyzing regional variations in resolution using the program Blocres (*Cardone et al., 2013*) indicates that the TM region is better resolved than the two NBDs (*Figure 1D*). The density map for ICP47 and the TMDs of TAP shows prominent side chain density sufficient to register the amino acid identity (*Figure 1A and C*). The density corresponding to the two NBDs is adequate to assign secondary structure but lacks side chain definition (*Figure 1*). With this map, we built and refined a model containing the 12 core TM-helices of TAP and the N-terminal 55 residues of ICP47 (*Table 2*). Poly-alanine models of the NBDs were generated from a rat TAP1 NBD crystal structure (PDB 1JJ7) (*Gaudet and Wiley, 2001*) using the program Modeller (*Webb and Sali, 2016*) and were placed into the cryo-EM density as rigid bodies. The molecular model was refined using only half of the data while keeping the other half as a free set for validation (*Figure 1E*).

## The structure of TAP trapped by a viral inhibitor

The TAP structure has the canonical fold of ABC exporters (*Figure 2A and B*). The 12 transmembrane (TM) helices at the core are arranged into two bundles, each consisting of TM 1–3 and 6 of one subunit and TM 4–5 of the other subunit (*Figure 2B*). The NBDs are attached to each TM bundle at their cytoplasmic ends and are separated from each other. The N-terminal 55 residues of ICP47 form a helical hairpin structure, which inserts into the opening between the two TM bundles. No density was observed for either the C-terminal 33 residues of ICP47 or the two TMD0s, indicating these regions are flexible in this conformation.

Although the overall conformation of TAP can be described as inward facing, such that the TM pathway is accessible from the cytosol, there is a unique feature that has not been observed in any other ABC transporters. In the inward-facing structures of other ABC exporters such as P-gp (pdb: 4F4C), access to the TM pathway is blocked from the extracellular side of the membrane by TM 1 and TM 6 of both subunits (*Figure 2C*) (*Jin et al., 2012*). In TAP, however, this gating region is cracked open by ICP47 to a width approximately 4 Å in diameter (*Figure 2C*). In addition, a lateral opening to the membrane leaflet is observed near the ER lumen side (*Figure 2D*). The presence of

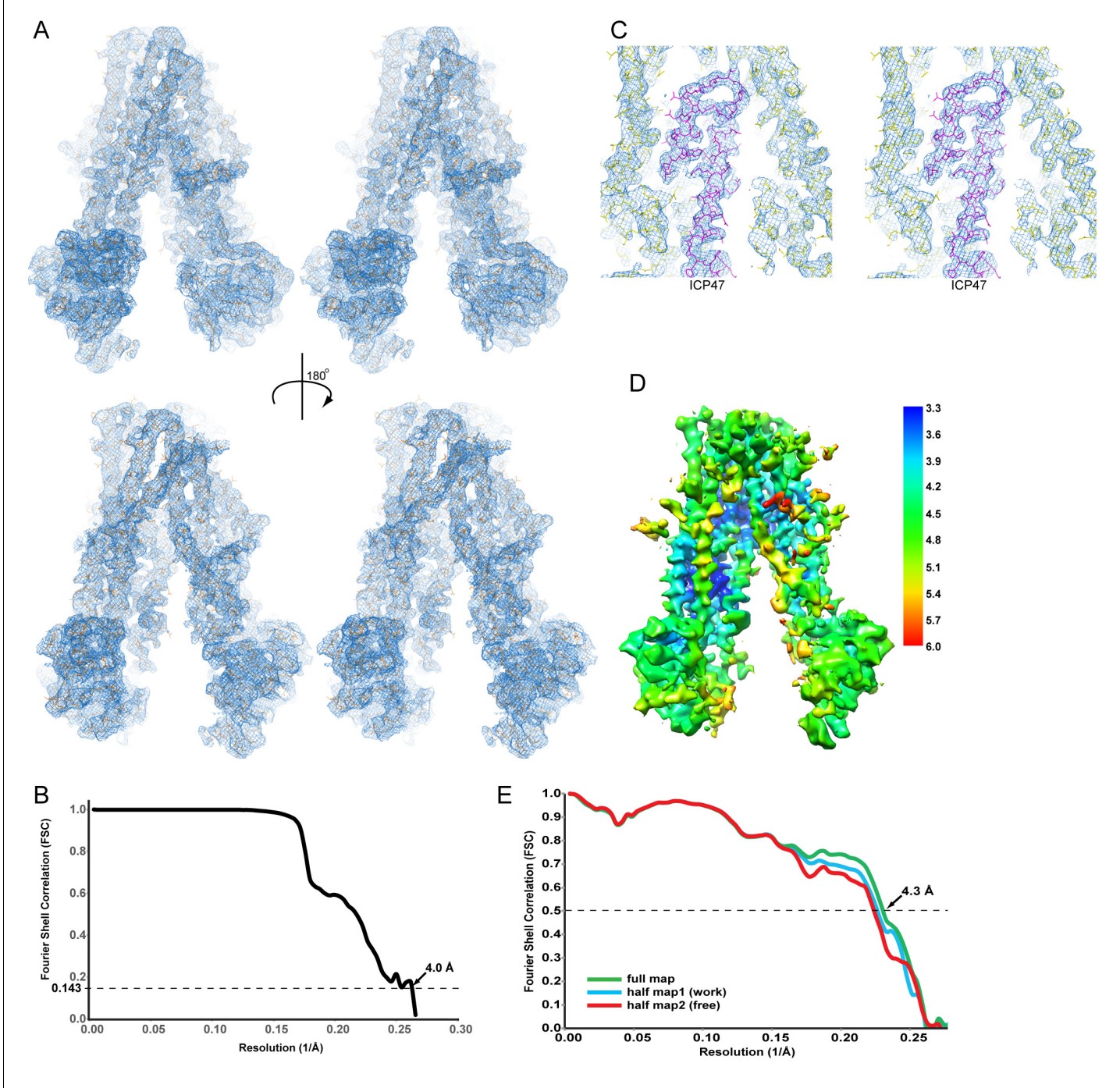

**Figure 1.** Cryo-EM reconstruction of the TAP/ICP47 complex. (**A**) Stereo views of the overall density map (blue mesh), filtered to 4 Å resolution and sharpened with a B-factor of −150 Å², for two 180° related views. The TAP/ICP47 model is shown in stick model (orange). (**B**) Resolution of the final cryo-EM density map indicated by a plot of the Fourier Shell Correlation (FSC) between unfiltered reconstructions of two semi-independently refined half datasets. (**C**) Stereo view of the density map (blue mesh) highlighting the TAP/ICP47 interface. Models of TAP (yellow) and ICP47 (magenta) are also shown. (**D**) Overall density map colored by local resolution estimation calculated from two semi-independently refined and reconstructed Frealign half maps using the Bsoft program Blocres and a 20 voxel kernel size. (**E**) Validation of the structure model. FSC calculated between the structure model and the half map used for refinement (working, cyan), the other half map (free, red), and 3) the full map (green).

The following source data is available for figure 1:

**Source data 1.** Resolution of the final cryo-EM reconstruction.

*Figure 1 continued*

**Source data 2.** Validation of the structure model.

ICP47 blocks both openings; otherwise there would be a continuous pathway across the membrane, a state violating the alternating access model (*Jardetzky, 1966*). Therefore, it is most likely that the ER openings observed in the structure of TAP/ICP47 complex are induced by ICP47. A functional TAP without inhibitor would presumably open the TM pathway to only one side of the membrane at a time to avoid potential ion leakage across the ER membrane.

## The peptide-binding pocket

Although we do not yet have direct structural data regarding where or how natural substrates bind, there is strong evidence suggesting that ICP47 competes with substrates for the same binding pocket (*Ahn et al., 1996*; *Früh et al., 1995*; *Hill et al., 1995*; *Tomazin et al., 1996*). Studies using radiolabeled peptides and mass spectrometry identified four regions, residues 375–420 and 453–487 in TAP1 and 301–389 and 414–433 in TAP2, as part of the binding site (*Nijenhuis and Hämmerling, 1996*). Mapping these residues onto the structure shows that they are part of the TM helices enclosing the large internal cavity (*Figure 3A*). The electrostatic environment of the TM cavity containing strong positively and negatively charged patches provides an appropriate interface for binding peptides with free N- and C-termini (*Figure 3B*) (*Momburg et al., 1994b*; *Schumacher et al., 1994*). Many residues identified by mutagenesis or substrate crosslinking experiments are located on the surface of this cavity (*Figure 3A and C*) (*Armandola et al., 1996*; *Deverson et al., 1998*; *Geng et al., 2015*; *Momburg et al., 1996*). In the TAP/ICP47 complex these residues are buried by ICP47, consistent with data showing that ICP47 precludes peptide binding (*Ahn et al., 1996*; *Früh et al., 1995*). Interestingly, the two residues proposed to serve as anchors for the C-termini of bound peptides, Y408 of TAP1 and M218 of TAP2 (*Geng et al., 2015*; *Momburg et al., 1996*), both interact with the N-terminal region of ICP47 through van der Waals contacts (*Figure 3D*). This observation is consistent with our previous suggestion that substrates interact with TAP through a very different mode from that of ICP47 (*Oldham et al., 2016*).

**Table 2.** Reciprocal space refinement statistics

| Space group | P1 |
| --- | --- |
| Cell dimensions | |
| a, b, c (Å) | 92.5, 116.0, 116.0 |
| α,β,γ (°) | 90.0, 90.0, 90.0 |
| Resolution (Å) | 100.0 - 3.97 |
| Number of residues | |
| TAP1 | 561 |
| TAP2 | 551 |
| ICP47 | 55 |
| R.m.s deviations | |
| Bond lengths (Å) | 0.0070 |
| Bond angles (°) | 0.881 |
| Ramachandran | |
| Favored (%) | 94.3 |
| Allowed (%) | 5.5 |
| Outliers (%) | 0.2 |

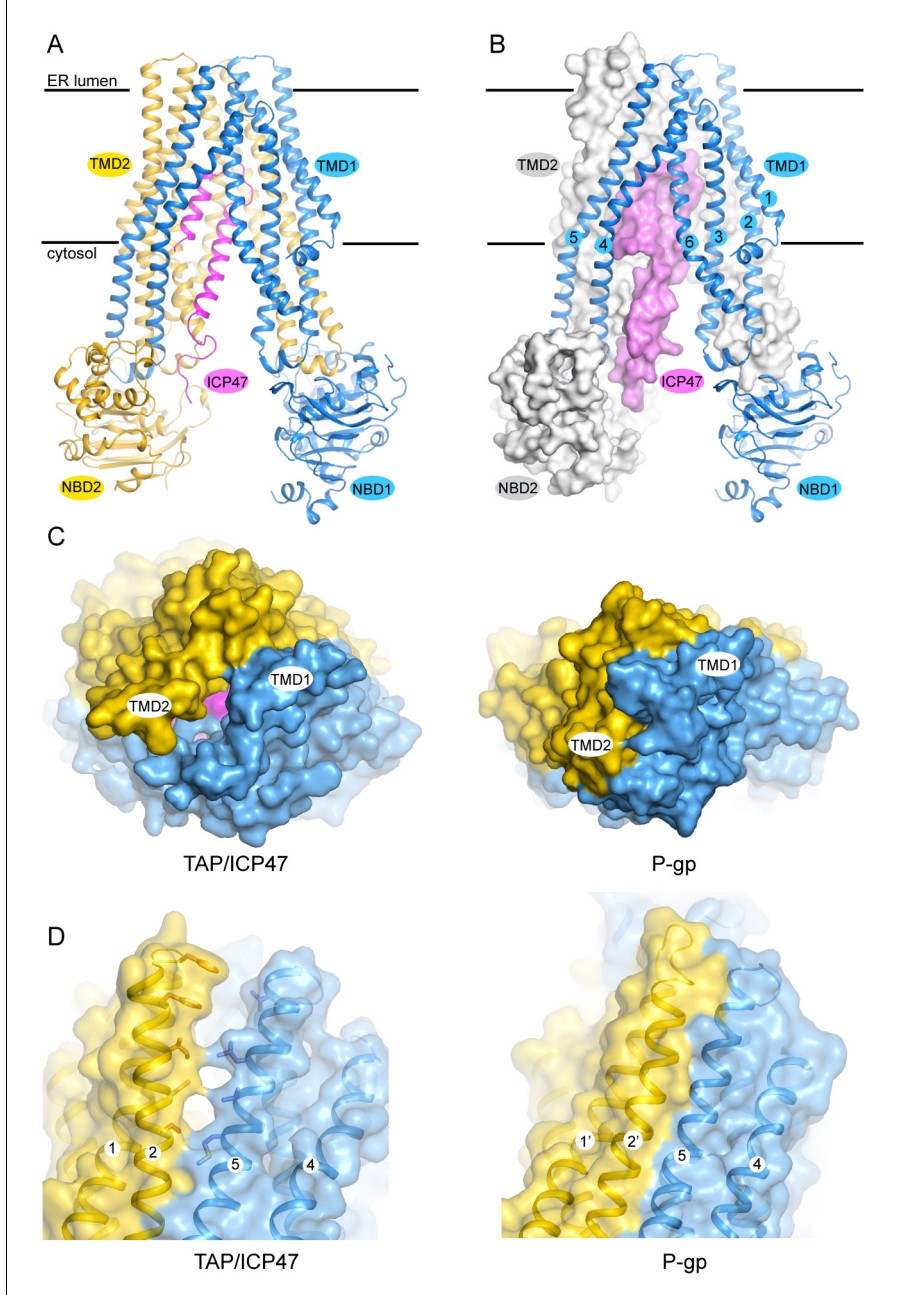

**Figure 2.** The structure of TAP trapped by ICP47. (**A**) Ribbon representation of the TAP/ICP47 complex. Color code: TAP1 (blue), TAP2 (yellow), ICP47 (magenta) (**B**) The domain-swapped architecture: TAP1 is shown in ribbon representation, TAP2 and ICP47 are shown as surfaces. TAP1 TM helices are labelled. (**C**) The open ER luminal gate viewed along the membrane normal from the ER side (left). The closed extracellular gate of P-gp is also shown for comparison (right). (**D**) The lateral opening to the membrane bilayer at the ER luminal side (left). The equivalent region in P-gp is also shown for comparison (right).

## The TAP/ICP47 interface

The interaction surface between ICP47 and TAP is extensive: 32 residues from ICP47 and 36 residues from TAP come into direct contact at the interface. The solvent-accessible surface area of TAP buried by ICP47 is 2360 Å$^2$ (*Figure 4A*), which is twice the average binding surface between proteins in

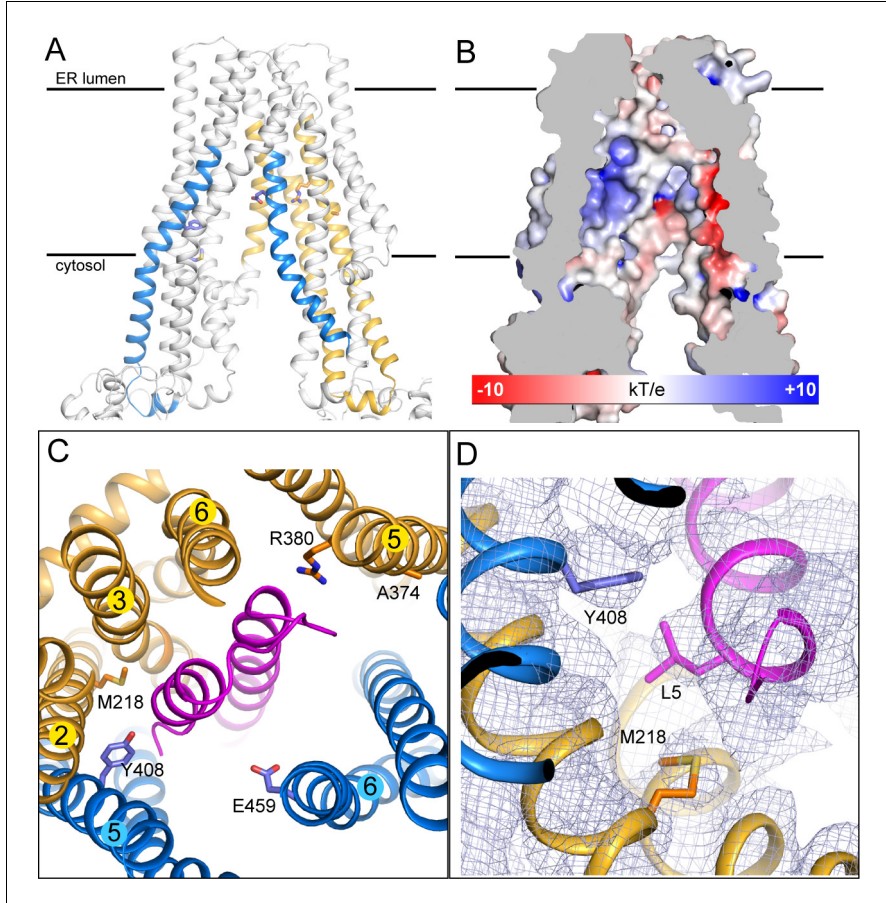

**Figure 3.** The substrate-binding site. (**A**) Biochemically identified substrate-binding regions: TAP1 375–420 and 453–487 (blue), TAP2 301–389 and 414–433 (gold). The five residues previously suggested to interact with the substrate (TAP1 Y408, E459 and TAP2 M218, A374, R380) are shown in stick model. (**B**) The electrostatic potential surface of the substrate-binding cavity. The electrostatic potential was calculated assuming pH 7 and a 0.15 M concentration of both (+1) and (−1) ions. Isocontour levels ranging from −10 to 10kT/e are colored from red to blue. (**C**) The helical hairpin of ICP47 (purple) plugs into the substrate-binding site. (**D**) The N-terminal region of ICP47 packs against Y408 of TAP1 and M218 of TAP2. For clarity, only side chains of TAP1 408, TAP2 M218, and ICP47 L5 are shown. The blue mesh shows the B-factor sharpened cryo-EM reconstruction.

general (*London et al., 2010*) and four times the surface area of an MHC-I molecule buried by a peptide antigen (*Rudolph et al., 2002*).

The N-terminal 34 residues of ICP47 are largely buried inside the translocation pathway, forming a helical bundle with TAP2 TM helices 3 and 6 (*Figure 3C* and *4C*). Previous studies suggest that this contact alone is sufficient to prevent substrate binding and subsequent conformational changes associated with transport (*Galocha et al., 1997*; *Neumann et al., 1997*). Residues 35–55 of ICP47 continue to pack closely along TAP2 TM3 and make contact with IH1 (intracellular helix 1), a coupling helix at the TMD/NBD interface. The last ordered residue of ICP47, P55, interacts with Y477, which normally makes an aromatic stacking interaction with the adenine ring of ATP (*Figure 4B*). The interactions between ICP47 and TAP1 are less extensive and are largely confined to the loop region at the tip of the helical hairpin.

A better understanding of which residues contribute most to the overall energy of binding comes from functional data, where each of the first 35 residues on ICP47 were mutated to alanine one at a time and assayed for TAP inhibition (*Galocha et al., 1997*). Mutations that reduced the activity by more than 50% are located in one region, from positions 18 to 25, where the two helices are connected by a sharp turn (*Figure 4C*). Previously, we generated a 'turn-to-helix' mutant by replacing

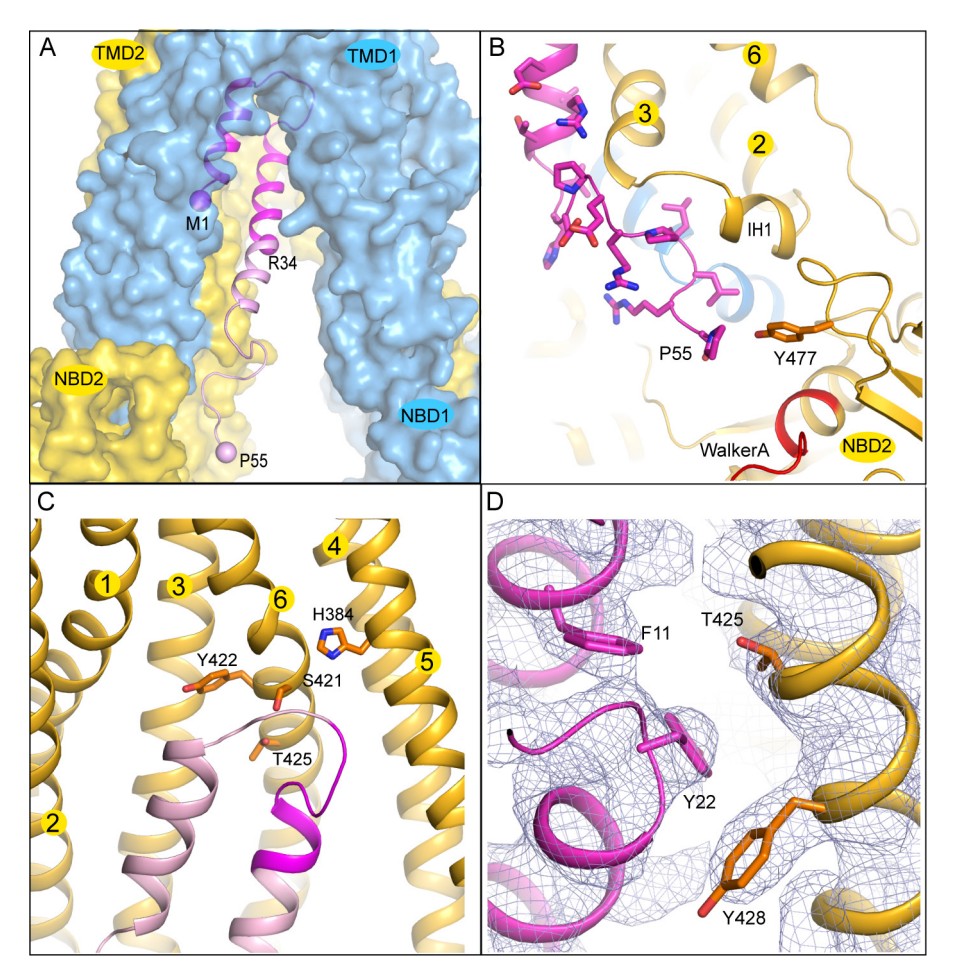

**Figure 4.** The interface between TAP and ICP47. (**A**) The first 34 residues of ICP47, highlighted in darker magenta, insert into the transmembrane pathway. R34, and the first and last residues of ICP47 resolved in the structure (M1, P55) are labeled. (**B**) ICP47 reaches into the TMD2/NBD2 interface near Y477. (**C**) Interactions between the 'hot-spots' in TAP shown in stick models and those of ICP47 (residues 18–25, highlighted in darker magenta). (**D**) Interactions between TAP2 T425 and ICP47 F11 and Y22. For clarity, only side chains of TAP2 T425 and Y428 and ICP47 F11 and Y22 are shown. The blue mesh shows the B-factor sharpened cryo-EM reconstruction.

residues in the same region (16–22) with alanine (*Oldham et al., 2016*). This mutant, predicted to have a higher propensity to form a long alpha-helix rather than the hairpin structure, indeed reduced the activity of ICP47 by a factor of five (*Oldham et al., 2016*).

To understand which residues on TAP are critical for binding to ICP47, we compared sequences of TAP that are sensitive to ICP47 inhibition with those resistant to it (*Figure 5*). Previous studies showed that ICP47 from HSV inhibits TAP in human, monkey, pig, cow and dog cells, but not in rabbit, mouse and rat cells (*Ahn et al., 1996*; *Jugovic et al., 1998*; *Verweij et al., 2011*). Among the 36 residues that make contacts with ICP47, a small set of residues are highly conserved in ICP47-sensitive species but not in rabbit, mouse and rat (*Figure 5*, boxed residues). When we map these residues onto the human TAP/ICP47 structure we observe that all but one of them are located near the apex of the TM cavity, interacting with the 'hot spot' residues in ICP47 that were identified in the alanine scan (*Figure 4C*) (*Galocha et al., 1997*). Thus, it appears that in the setting of a very large interface between TAP and ICP47, amino acid changes in this region are particularly important and account for TAP susceptibility to ICP47 among the species analyzed here. For example, both mouse and rat contain a T425N substitution, which is incompatible with the close contacts between T425

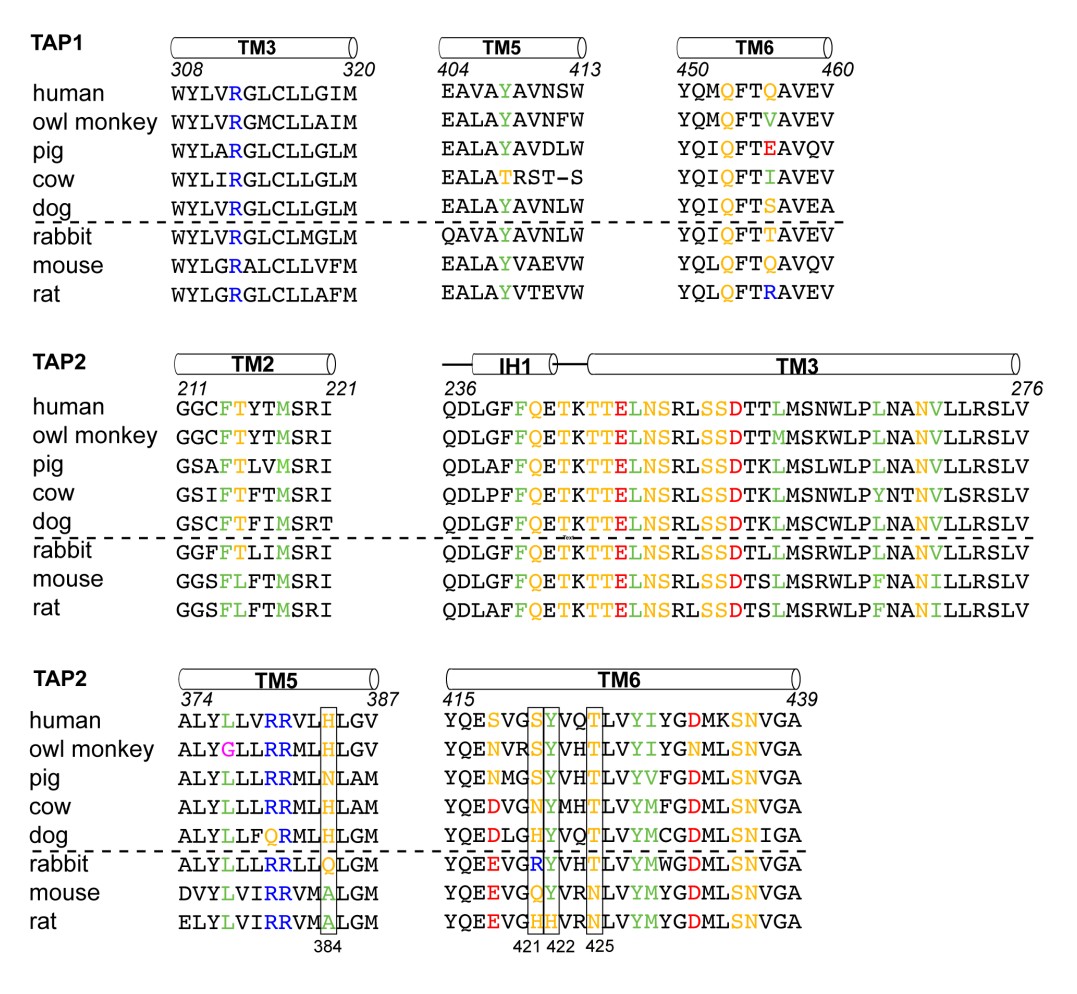

**Figure 5.** Sequence alignment of the TAP residues that contact ICP47. ICP47 inhibits TAP from human, owl monkey, pig, cow and dog (the top five sequences), but not that of rabbit, mouse and rat (the bottom three sequences). Residues contacting ICP47 are colored based on amino acid character (positively charged in blue, negatively charged in red, hydrophobic in green, glycine in magenta, and polar in gold). Residues discussed in the text are highlighted in boxes.

and F11 and Y22 of ICP47 (*Figure 4D*). Substituting F11 or Y22 with alanine reduced ICP47 inhibition by 40% and 60% respectively (*Galocha et al., 1997*). The lack of ICP47 inhibition in rabbit cells can be explained by the substitution of S421 with the large, charged arginine residue, which probably prevents insertion of the helical hairpin into the top of TM cavity (*Figure 4C*). Therefore, two independent approaches—mutagenesis to identify functional hotspots on ICP47 (*Galocha et al., 1997*) and analysis of specificity determinants on TAP—both point to the same molecular interface at the tip of the helical hairpin as being crucial to the action of ICP47 to inhibit TAP (*Figure 4C*).

## A mechanism of immune evasion by HSV

The structure of the TAP/ICP47 complex provides us with a clear picture of how HSV evades immune surveillance. TAP transports peptide antigens into the ER through conformational changes powered by ATP binding and hydrolysis. Based on what we have learned from homologous ABC transporters, we can envision that in the absence of substrate and ATP, TAP rests in an inward-facing conformation in which the two NBDs are separated and the translocation pathway faces the cytoplasm. Upon binding to the substrate and ATP, the two NBDs form a closed dimer and the translocation pathway orients towards the ER lumen to release the peptide. ATP hydrolysis at the closed NBD dimer interface resets TAP to its resting state, ready for the next transport cycle. ICP47 precludes substrate

binding by inserting a long helical hairpin into the translocation pathway. The strong interaction between ICP47 and TAP traps TAP in an inward-facing conformation with the two NBDs separated, unable to progress to the NBD-closed conformation necessary for ATP hydrolysis. By blocking translocation of viral peptides into the ER, HSV suppresses the MHC-I antigen presentation pathway and thereby escapes cytotoxic T cell detection.

## Structural comparison of peptide transporters in the ABC family

Previously, crystal structures were determined for two prokaryotic peptide transporters: the *E. coli* McjD in the AMPPNP-bound conformation (pdb: 4PL0) (*Choudhury et al., 2014*; *Mehmood et al., 2016*) and the peptidase-containing ABC transporter from *Clostridium thermocellum* (PCAT1) in two different conformations (inward-facing pdb: 4RY2; occluded, ATPγS-bound pdb: 4S0F) (*Lin et al., 2015*). Unlike TAP, which is a promiscuous transporter found only in jawed vertebrates (*Hinz et al., 2014*), McjD and PCAT1 are dedicated to specific substrates. McjD exports microcin J25, a 21-residue antibacterial peptide with a lasso fold. PCAT1 functions both as a maturation protease and exporter for a 90-residue peptide with an amino-terminal leader sequence. Correspondingly, PCAT1 contains two peptidase domains in addition to the canonical TMDs and NBDs. Despite these differences, the three transporters share a similar fold in the core region: the NBDs have very similar

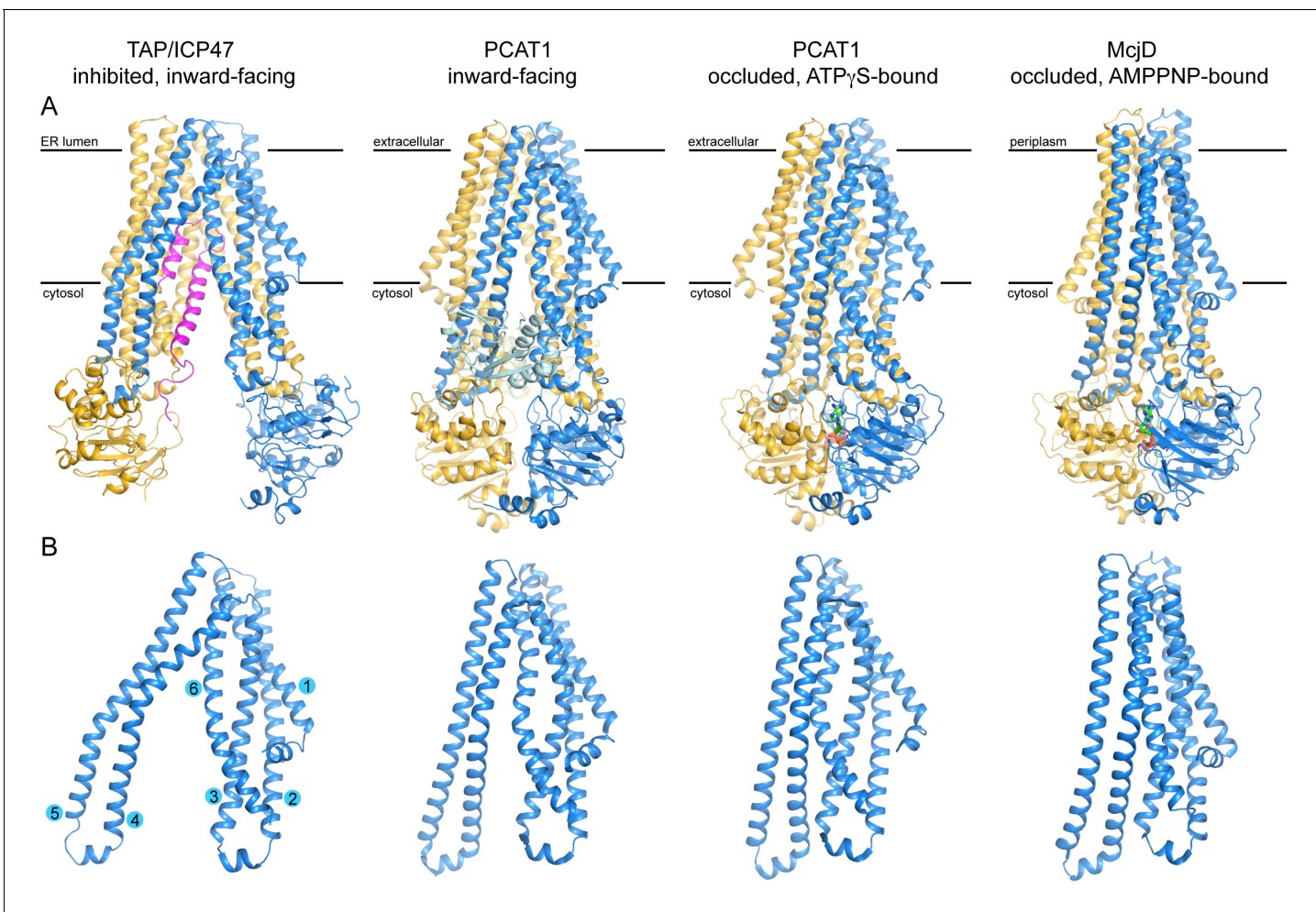

**Figure 6.** Structures of three peptide transporters in the ABC family. (**A**) Ribbon representations. The two subunits are shown in blue and gold, respectively. ICP47 is colored in magenta. The peptidase domains of PCAT1 observed in the inward-facing conformation are colored in cyan and light yellow. The nucleotides, ATPγS in PCAT1 and AMPPNP in McjD, are shown in stick model. (**B**) Structural comparison of the TMDs. Only one TMD is shown for each transporter. The TM helices for TAP1 are shown.

structures and the TM helices in each subunit are of similar length with a similar trajectory (*Figure 6*). One notable difference between TAP and the other two transporters occurs in TM 4-5, which in TAP appears to be pushed outward and bent (*Figure 6*). This difference, however, could possibly reflect a distortion due to the presence of ICP47. The overall structures of TAP, PCAT1, and McjD differ in the separation of the two halves, reflecting the different conformational states each structure represents. TAP shows the largest separation, again possibly induced by the viral inhibitor ICP47. The two NBDs of PCAT1 form a semi-open dimer in the absence of a nucleotide, separated at the TMD/NBD interfaces and making contacts at the distal end of the structure (*Figure 6*). In the presence of ATP analogs and the absence of a substrate, both McjD and PCAT1 reveal an occluded conformation in which the NBDs are closed and the translocation pathway is shielded from both sides of the membrane (*Figure 6*). The similarity in their structures suggests that these transporters may share a common evolutionary origin and a common mechanism for coupling ATP hydrolysis to peptide translocation.

## Materials and methods

### Expression of TAP and ICP47 and co-purification of the TAP/ICP47 complex

Human TAP and HSV-1 ICP47 were expressed and purified as described in the earlier study (*Oldham et al., 2016*). Briefly, ICP47 was expressed in *E. coli* and purified via a N-terminal glutathione S-transferase (GST) affinity tag. *Pichia pastoris* cells (strain SMD 1163 His+; Invitrogen) co-expressing TAP1 and TAP2 were lysed with a mixer miller (Retsch Mixer Mill 400) and incubated with purified ICP47 before solubilizing with n-Dodecyl β-D-maltoside (DDM; Anatrace). The TAP/ICP47 complex was purified on IgG Sepharose resin (GE Healthcare) via the Protein A tag at the C-terminus of TAP1. The Protein A tag was removed by PreScission protease and the complex was further purified using a Superose 6 column (GE Healthcare) in a buffer containing 20 mM Hepes, pH 7.4, 150 mM NaCl, 2 mM TCEP, 1 mM DDM, and 1 mM octaethylene glycol monododecyl ether (C12E8; Anatrace).

### Electron microscopy sample preparation and microscope imaging

Cryo-EM grids were prepared as described (*Oldham et al., 2016*). Briefly, 3 µl of purified TAP/ICP47 complex (2 mg/ml) was pipetted onto glow-discharged C-flat holey carbon CF-1.2/1.3–4C grids (Protochips). At 90% humidity, the grids were blotted for 4 s using a Vitrobot Mark IV (FEI) and frozen in liquid ethane. Imaging data were collected on a FEI Titan Krios electron microscope (acceleration voltage of 300 keV) with a K2 Summit direct electron detector (Gatan Inc.) running in super-resolution counting mode and using SerialEM (*Mastronarde, 2005*). A Gatan Imaging filter with a slit width of 10 eV was used to remove inelastically scattered electrons. Movie frames were recorded on a single grid with a total exposure time of 10 s (200 ms per frame) using a dose rate of 8 electrons/pixel/s or 7.4 electrons/Å$^2$/s.

### Image processing

Movie frames were corrected using a gain reference and binned by a factor of 2, resulting in a pixel size of 1.04 Å. The effective contrast transfer function (CTF) was determined from the frame-summed micrographs using CTFFIND4 (*Rohou and Grigorieff, 2015*). Manual picking and 2D classification was performed in Relion to produce template classes for autopicking (*Scheres, 2012*). Particles automatically selected by Relion were inspected manually to remove false positives, resulting in a dataset of about 502,000 particles.

For specimen movement correction, we compared the results from three different methods: whole frame alignment using Unblur (*Grant and Grigorieff, 2015*), individual particle alignment using alignparts_lmbfgs (*Rubinstein and Brubaker, 2015*), and by first aligning frames with Unblur then aligning individual particles in the Unblur-aligned movies using alignparts_lmbfgs. Using these three different procedures, the best resolution values obtained at the stage of AutoRefine3D in Relion were 7.5 Å, 7.4 Å, and 6.6 Å, respectively. Thus, the best results were obtained by combining whole frame alignment with subsequent individual particle tracking.

Final reconstruction and refinements were carried out in Frealign (*Grigorieff, 2016*) using particles aligned with Unblur and alignparts_Imbfgs. Global parameter search (mode 3) was performed at 8.0 Å resolution, followed by several iterations of local refinement with the alignment resolution limit gradually increasing from 8.0 to 6.0 Å (mode 1). The resolution of the final reconstruction was estimated at 4.0 Å using the Fourier shell correlation (FSC) of two reconstructions each containing half of the data and using 0.143 as the cut-off criterion (*Figure 1B*).

## Model building

A model, consisting of residues 173–742 of TAP1, residues 130–681 of TAP2, and residues 1–55 of ICP47, was manually built in Coot (*Emsley et al., 2010*). Several regions, including TAP1 residues 173–183, 215–222, 272–282, 322–325, 336–347, 431–443 and TAP2 residues 181–186 have poor density and were registered based on the homologous structure ABCB10 (PDBcode 4AYT) (*Shintre et al., 2013*).

## Refinement and validation

Model refinement was performed in both real and reciprocal space. Using the program Pdbset (*Winn et al., 2011*), the TAP/ICP47 model was translated into a P1 crystallographic symmetry unit cell which was padded by 5 Å in each axis. The full map and the two half maps from Frealign were also translated into the unit cell using the program Maprot (*Stein et al., 1994*). To generate a working half map for refinement, structure factors and phases were calculated from one of the translated half maps using the program Sfall (*Ten Eyck, 1977*). The model was then refined against the working half map using PHENIX real space refine with secondary structure restraints imposed (*Adams et al., 2010*). Subsequently, the structure was refined against the working half map in reciprocal space using Refmac (*Brown et al., 2015*; *Murshudov et al., 1997*) with secondary structure restraints calculated from ProSMART (*Nicholls et al., 2014*). We used the EMAN2 program suite (*Tang et al., 2007*) to produce map from the atomic coordinates of the complex model. To access the degree of overfitting, we calculated FSC curves between the model and the half map used for refinement (work), the other half-map (free), and the full map (*Figure 1*). The FSC curves were calculated using Spider (*Frank et al., 1996*) by resampling the model map onto the same grid as the data maps using UCSF Chimera (*Pettersen et al., 2004*) and calculating FSC curves between this converted map and the cryo-EM maps. The cryo-EM maps were masked using a generous mask with a smooth edge and a volume exceeding the estimated volume of the model by about 3.5 times. The FSC curves were then adjusted for the volume exceeding the volume of the model using the formula

$$\mathrm{FSC_{corrected}} = f * \mathrm{FSC} / (1 + (f-1) * \mathrm{FSC})$$

where $f$ is the factor by which the mask exceeds the volume of the model (*Sindelar and Grigorieff, 2012*). The FSC curve (green) between the model and the full map has a value of 0.5 at a resolution of 4.3 (*Figure 1E*).

## Figure preparation

Figures were prepared using the programs PyMOL (*Schrödinger LLC, 2015*) and UCSF Chimera (*Pettersen et al., 2004*).

## Acknowledgements

We thank Mark Ebrahim and Johanna Sotiris at the Rockefeller Evelyn Gruss Lipper Cryo-Electron Microscopy Resource Center and Zhiheng Yu, Chuan Hong and Rick Huang at the Janelia Research Campus Hughes Medical Institute cryo-EM facility for assistance in data collection. We also thank Timothy Grant and Alexis Rohou in the Grigorieff laboratory for advice in image processing, and Sarah McCarry, Jonathan Whicher and Anthony Palillo for editing this manuscript. We thank the Janelia Visiting Scientist Program for supporting JC's sabbatical visit to NG. JC is an Investigator of the Howard Hughes Medical Institute. The 3D cryo-EM TAP ICP47 density map has been deposited in the Electron Microscopy Data Bank under the accession number EMD-8482. Coordinates of the built TAP ICP47 structure have been deposited in the Protein Data Bank with accession code 5U1D.

## Additional information

### Competing interests

NG: Reviewing editor, *eLife*. The other authors declare that no competing interests exist.

### Funding

| Funder | Author |
| --- | --- |
| Howard Hughes Medical Institute | Nikolaus Grigorieff |
| | Jue Chen |

The funders had no role in study design, data collection and interpretation, or the decision to submit the work for publication.

### Author contributions

MLO, JC, Conception and design, Acquisition of data, Analysis and interpretation of data, Drafting or revising the article; NG, Conception and design, Analysis and interpretation of data, Drafting or revising the article

### Author ORCIDs

Nikolaus Grigorieff, http://orcid.org/0000-0002-1506-909X
Jue Chen, http://orcid.org/0000-0003-2075-4283

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
