## [Decision Letter]

Thank you for submitting your article "Atomic structure of the transporter associated with antigen processing trapped by herpes simplex virus" for consideration by *eLife*. Your article has been favorably evaluated by Richard Aldrich (Senior Editor) and three reviewers, one of whom is a member of our Board of Reviewing Editors. The following individual involved in review of your submission has agreed to reveal their identity: Wesley I Sundquist (Reviewer #2).

The reviewers have discussed the reviews with one another and the Reviewing Editor has drafted this decision to help you prepare a revised submission.

Summary:

The authors describe a 4Å cryo-EM reconstruction of the complex formed by the heterodimeric TAP transporter and the Herpesviral TAP inhibitor ICP47. The work builds on the previous 6.5Å structure reported by this same group in Nature last January. The enhanced resolution was made possible by collecting data at higher magnification (to improve alignment), optimizing other aspects of data collection (e.g., collecting images from regions of thin ice), and optimized data processing (e.g., in improved movie alignment protocols). The advance is that an atomic model can now be built for the core of the complex, which enables the authors to identify intermolecular contacts and interpret the structure in terms of comparative sequence analyses and previous mutational analyses of TAP and ICP47.

Essential revisions:

The reviewers have discussed the manuscript focusing on the balance of the obviously important achievement of resolving the structure at 4 Å resolution, which allows the residue-specific model building, and somewhat limited mechanistic insights provided by the manuscript.

The following are comments to be addressed:

1) The authors should provide a bit better overview of the complex in the current manuscript so that it "stands alone" a bit better. For example, it would be helpful to give readers a better overview of how ICP47 binding alters the nucleotide binding cleft. The inhibition of ATPase activity is implied in the second paragraph of the subsection “The TAP/ICP47 interface”, but the actual bound nucleotide state (apo if I understand correctly) and an explanation of how NBD closure is prevented by ICP47 binding are not really addressed as far as I can see (though this issue was mentioned in the Nature paper).

2) I would have also liked a more extensive comparison with the homologous transporters.

3) It would be helpful if data collection strategy were further discussed and illustrated by figures. For example, quantitative evaluation of different methods to correct for specimen movements would have been informative.

4) The authors indicate that collecting data from areas of the grid with good contrast (presumably where the ice was thinnest) contributed to the improved resolution. This seems logical, but can the authors quantify how the selected regions were chosen (so that others can follow a similar protocol) and/or quantify how much this selection contributed to the increased resolution?

5) The number of particles (502K, minus the 5% that received such low Frealign scores that they were effectively eliminated) is mentioned in the text, but not in either of the two data tables.

6) A structure at 4 Å resolution cannot – and should not – be described as "atomic". This is repeatedly and prominently the case, e.g. in the Abstract and summary. The inflationary use of the term "atomic" (structure, resolution, model, interactions etc.) in connection with lowish-resolution cryoEM maps is seriously misleading and has to be avoided ("near-atomic" is more appropriate and now widely used and accepted). Anyone can build an atomic model into a map of any resolution, and describe the result as an "atomic structure". What is the point?

7) It is also potentially misleading and not to be encouraged to show van der Waals surfaces for interacting residues, instead of the genuine cryoEM map, as in Figure 3 and Figure 4. If the map is good enough, why not show it?

8) The important interaction of ICP Pro55 with Tyr477 of TAP2 is not shown. Pro55 is modelled as a pink sphere, which seems to be too far from Tyr477 for direct, close interaction that would displace an ATP bound by strong π stacking.

9) I assume the structure is that of the human transporter. The manuscript does not state this important fact explicitly anywhere.

---

## [Author Response]

Essential revisions:

The reviewers have discussed the manuscript focusing on the balance of the obviously important achievement of resolving the structure at 4 Å resolution, which allows the residue-specific model building, and somewhat limited mechanistic insights provided by the manuscript.

The following are comments to be addressed:

1) The authors should provide a bit better overview of the complex in the current manuscript so that it "stands alone" a bit better. For example, it would be helpful to give readers a better overview of how ICP47 binding alters the nucleotide binding cleft. The inhibition of ATPase activity is implied in the second paragraph of the subsection “The TAP/ICP47 interface”, but the actual bound nucleotide state (apo if I understand correctly) and an explanation of how NBD closure is prevented by ICP47 binding are not really addressed as far as I can see (though this issue was mentioned in the Nature paper).

In the revision, we added a section entitled “A mechanism of immune evasion by HSV”.

2) I would have also liked a more extensive comparison with the homologous transporters.

In the revision, we added a section entitled “Structural comparison of peptide transporters in the ABC family” and a new figure (Figure 6) to accompany the discussion.

3) It would be helpful if data collection strategy were further discussed and illustrated by figures. For example, quantitative evaluation of different methods to correct for specimen movements would have been informative.

We now included in the subsection “Electron microscopy sample preparation and microscope imaging”, the details of how different movie processing procedures were carried out and quantified.

4) The authors indicate that collecting data from areas of the grid with good contrast (presumably where the ice was thinnest) contributed to the improved resolution. This seems logical, but can the authors quantify how the selected regions were chosen (so that others can follow a similar protocol) and/or quantify how much this selection contributed to the increased resolution?

The areas were selected manually, based on the experience we have gathered over the past two years working on this project. It is not possible to quantitatively describe this process as it was done by visual inspection.

5) The number of particles (502K, minus the 5% that received such low Frealign scores that they were effectively eliminated) is mentioned in the text, but not in either of the two data tables.

We have now included the number of particles in Table 1. We list the total number (502k) since the 5% of particles that did not contribute at 4 Å resolution still contributed at lower resolution.

6) A structure at 4 Å resolution cannot – and should not – be described as "atomic". This is repeatedly and prominently the case, e.g. in the Abstract and summary. The inflationary use of the term "atomic" (structure, resolution, model, interactions etc.) in connection with lowish-resolution cryoEM maps is seriously misleading and has to be avoided ("near-atomic" is more appropriate and now widely used and accepted). Anyone can build an atomic model into a map of any resolution, and describe the result as an "atomic structure". What is the point?

We have removed the term “atomic” as requested.

7) It is also potentially misleading and not to be encouraged to show van der Waals surfaces for interacting residues, instead of the genuine cryoEM map, as in Figure 3 and Figure 4. If the map is good enough, why not show it?

We modified Figure 3 and Figure 4 to replace the van der Waals surfaces with the cryoEM map as requested.

8) The important interaction of ICP Pro55 with Tyr477 of TAP2 is not shown. Pro55 is modelled as a pink sphere, which seems to be too far from Tyr477 for direct, close interaction that would displace an ATP bound by strong π stacking.

We have modified Figure 4 to show the details of this interaction.

9) I assume the structure is that of the human transporter. The manuscript does not state this important fact explicitly anywhere.

We have now stated in the text that the cryo-EM reconstruction is of the human TAP/ICP47 complex.